# One-Year Outcomes after Surgical versus Transcatheter Aortic Valve Replacement with Newer Generation Devices

**DOI:** 10.3390/jcm10163703

**Published:** 2021-08-20

**Authors:** Stefano Rosato, Fausto Biancari, Paola D’Errigo, Marco Barbanti, Giuseppe Tarantini, Francesco Bedogni, Marco Ranucci, Giuliano Costa, Tatu Juvonen, Gian Paolo Ussia, Andrea Marcellusi, Giovanni Baglio, Stefano Domenico Cicala, Gabriella Badoni, Fulvia Seccareccia, Corrado Tamburino

**Affiliations:** 1National Centre for Global Health, Istituto Superiore di Sanità, 00161 Rome, Italy; stefano.rosato@iss.it (S.R.); paola.derrigo@iss.it (P.D.); gabriella.badoni@iss.it (G.B.); fulvia.seccareccia@iss.it (F.S.); 2Clinica Montevergine, GVM Care and Research, 83013 Mercogliano, Italy; 3Division of Cardiology, A.O.U. Policlinico “G. Rodolico—San Marco”, University of Catania, 95124 Catania, Italy; mbarbanti83@gmail.com (M.B.); giulianocosta90@gmail.com (G.C.); tambucor@unict.it (C.T.); 4Division of Cardiology, Department of Cardiac, Thoracic and Vascular Sciences, University of Padova, 35122 Padova, Italy; giuseppe.tarantini.1@unipd.it; 5Department of Clinical and Interventional Cardiology, IRCCS Policlinico San Donato, San Donato Milanese, 20097 Milan, Italy; francesco.bedogni@grupposandonato.it; 6Department of Cardiothoracic and Vascular Anesthesia and ICU, IRCCS Policlinico San Donato, San Donato Milanese, 20097 Milan, Italy; cardioanestesia@virgilio.it; 7Heart and Lung Center, Helsinki University Hospital, University of Helsinki, 00029 Helsinki, Finland; tatu.juvonen@hus.fi; 8Research Unit of Surgery, Anesthesiology and Critical Care, University of Oulu, 90570 Oulu, Finland; 9Department of Interventional Cardiology, Campus Bio-Medico University of Rome, 00128 Rome, Italy; g.ussia@unicampus.it; 10Economic Evaluation and HTA (EEHTA), CEIS Faculty of Economics, University of Rome “Tor Vergata”, 00133 Rome, Italy; andrea.marcellusi@gmail.com; 11Italian National Agency for Regional Healthcare Services, 00187 Rome, Italy; baglio@agenas.it (G.B.); cicala@agenas.it (S.D.C.)

**Keywords:** transcatheter aortic valve replacement (TAVR), transcatheter aortic valve implantation (TAVI), aortic valve replacement

## Abstract

The superiority of transcatheter (TAVR) over surgical aortic valve replacement (SAVR) for severe aortic stenosis (AS) has not been fully demonstrated in a real-world setting. This prospective study included 5706 AS patients who underwent SAVR from 2010 to 2012 and 2989 AS patients who underwent TAVR from 2017 to 2018 from the prospective multicenter observational studies OBSERVANT I and II. Early adverse events as well as all-cause mortality, major adverse cardiac and cerebrovascular events (MACCEs), and hospital readmission due to heart failure at 1-year were investigated. Among 1008 propensity score matched pairs, TAVR was associated with significantly lower 30-day mortality (1.8 vs. 3.5%, *p* = 0.020), stroke (0.8 vs. 2.3%, *p* = 0.005), and acute kidney injury (0.6 vs. 8.2%, *p* < 0.001) compared to SAVR. Moderate-to-severe paravalvular regurgitation (5.9 vs. 2.0%, *p* < 0.001) and permanent pacemaker implantation (13.8 vs. 3.3%, *p* < 0.001) were more frequent after TAVR. At 1-year, TAVR was associated with lower risk of all-cause mortality (7.9 vs. 11.5%, *p* = 0.006), MACCE (12.0 vs. 15.8%, *p* = 0.011), readmission due to heart failure (10.8 vs. 15.9%, *p* < 0.001), and stroke (3.2 vs. 5.1%, *p* = 0.033) compared to SAVR. TAVR reduced 1-year mortality in the subgroups of patients aged 80 years or older (HR 0.49, 95% CI 0.33–0.71), in females (HR 0.57, 0.38–0.85), and among patients with EuroSCORE II ≥ 4.0% (HR 0.48, 95% CI 0.32–0.71). In a real-world setting, TAVR using new-generation devices was associated with lower rates of adverse events up to 1-year follow-up compared to SAVR.

## 1. Introduction

Transcatheter aortic valve replacement (TAVR) is slowly gaining acceptance for the treatment of lower risk patients with aortic valve stenosis (AS) after several randomized and observational studies confirmed comparable early results to surgical aortic valve replacement (SAVR) [1,2,3]. TAVR has been shown to be feasible also in challenging anatomic conditions such as in stenotic bicuspid aortic valve [4]. Randomized clinical trials provided evidence on the early and mid-term efficacy and safety of TAVR in AS patients within a broad spectrum of operative risk. However, clinical trials might have excluded a large number of patients who still undergo invasive treatment for severe AS in the real-world setting. Indeed, recent studies excluded from randomization up to one third of screened AS patients [1,2]. Large multicenter studies showed that SAVR can be performed with comparable mid-term outcomes to TAVR also in the very elderly [5]. Consonant with these studies, SAVR is still largely used in intermediate- and high-risk patients and very elderlies despite its invasive nature [6,7]. Comparative analyses often included patients who received older generation TAVR devices, which prevented a reliable assessment of the outcome with newer TAVR devices. In this controversial scenario, there is a need for data from large clinical registries to demonstrate the efficacy and safety of TAVR in the real-world setting when new devices were used. The aim of the present study was to compare the early and 1-year outcome of newer generation TAVR devices to SAVR in all-comers included in two national prospective studies.

## 2. Materials and Methods

### 2.1. Data Source

Data for the present analysis was gathered from merging the Observational Study of Effectiveness of SAVR–TAVI Procedures for Severe Aortic Stenosis Treatment (OBSERVANT) I and II datasets. OBSERVANT I was a national, observational, prospective, multicenter cohort study that enrolled consecutive AS patients who underwent TAVR or SAVR at 93 Italian centers (34 cardiology centers and 59 cardiac surgery centers) between December 2010 and June 2012 [8]. Hospitals participating in this study were able to offer either SAVR or TAVR for severe AS.

OBSERVANT II was a national observational, prospective, multicenter cohort study that enrolled consecutive AS patients who underwent TAVR at 30 Italian centers of cardiology between December 2016 and September 2018. Only 28 centers met the minimum data quality criteria required by the study protocol and their data is included in this analysis [9]. The Ethical Committees of the ASL Milano 2 (approval code 2574; date of approval 17 June 2010) and of the San Raffaele Hospital, Milan (approval code 126/2016; date of approval 5 May 2016) approved the OBSERVANT I and II studies, respectively. Each participating center was granted permission to participate in the OBSERVANT I and II studies. All patients included in these studies gave informed consent to the scientific treatment of their data anonymously.

Data on baseline characteristics, operative details, and adverse events occurred during the index hospitalization were prospectively collected into an electronic case report form. Data on adverse events occurred after hospital discharge was gathered by a linkage with the National Hospital Discharged Records database and the Tax Registry Information System, provided by the Italian Ministry of Health through a collaboration with the Italian National Program for Outcome Evaluation (PNE-AGENAS). Linking to these national registries guaranteed complete follow-up data on outcomes at 1-year follow-up. This study was performed following the STrengthening the Reporting of OBservational studies in Epidemiology (STROBE) guidelines (Appendix A) [10]. 

### 2.2. Study Population

The study population of this study consisted of patients with severe AS who underwent SAVR or TAVR with or without coronary revascularization. Age < 30 years, older generation TAVR devices, emergency procedure, active endocarditis, porcelain aorta, hostile chest and severe frailty as defined by Geriatric Status Scale grade 3 [11] were the exclusion criteria for this analysis.

### 2.3. Outcomes

The primary outcomes of interest were all-cause mortality, major adverse cardiac and cerebrovascular events (MACCEs), and hospital readmission due to heart failure at 1-year. Secondary outcomes were the following adverse events occurring during the index hospitalization: stroke, conversion to cardiac surgery, complication at the left ventricular apex, major vascular injury, acute kidney injury, postoperative change in estimated glomerular filtration rate, permanent pacemaker implantation, cardiogenic shock, infection complications and its sites, red blood cell transfusion, procedure for cardiac tamponade, mean and peak transvalvular gradient after the procedure and paravalvular regurgitation [12]. Thirty-day death was among the secondary early outcomes. Secondary late outcomes were also the following adverse events, which occurred at 1-year: reoperation for aortic valve prosthesis complications, permanent pacemaker implantation, stroke, myocardial infarction, percutaneous coronary intervention, and coronary artery bypass grafting.

MACCE was defined as a composite end-point including any of the following adverse events: all-cause mortality, stroke, myocardial infarction, and coronary revascularization. For the purpose of this study, major vascular injury was defined as any vascular complication at the peripheral access site requiring surgical or endovascular intervention. Among the major vascular complications herein considered, we included major injuries of the aorta such as aortic dissection, whilst injury at the left ventricular access site was excluded from this category. Infectious complications were defined as clinically proven surgical site infections, infections involving organs or spaces, and sepsis.

### 2.4. Statistical Analysis

Continuous variables are reported as means and standard deviations. Categorical variables are reported as counts and percentages. Missing data were not replaced. The Mann–Whitney *U* test, the Fisher exact test, and the χ^2^ test were used for univariate analysis in the unmatched population. A propensity score was estimated using a non-parsimonious logistic regression model, including all covariates listed in Table 1 except for hemoglobin and variables related to the aortic valve because of missing data. One-to-one propensity score matching was performed using the nearest-neighbor method and a caliper width of 0.2 of the standard deviation of the propensity score logit. The *t* test for paired samples for continuous variables, the McNemar test for dichotomous variables, and the analysis of the standardized differences were used to evaluate the balance between the matched groups. Standardized differences less than 0.10 were considered an acceptable imbalance between the study groups. Early outcomes in the propensity score matched cohorts were evaluated using the *t* test for paired samples for continuous variables and the McNemar test for dichotomous variables. Differences in the long-term survival and MACCEs of matched pairs were evaluated using the Kaplan–Meier method with the Klein–Moeschberger stratified log-rank test. Analysis of non-fatal adverse events was performed using the Fine-Grays method considering all-cause death as a competing event. The previously described propensity score matching was implemented to rematch on the following subgroups of patients ignoring the match on full cohort: females versus males, patients older and younger than 80 years, with or without coronary artery disease, with EuroSCORE II below or higher than 4.0%, left ventricular ejection fraction below or higher than 50%, and with or without diabetes. These matched data sets were used for interaction tests analyses. Risk estimates are reported as hazard ratios (HR) and subdistributional hazard ratios (SHR) with their 95% confidence interval (CI). *p*-values were 2-tailed. *p* < 0.10 was considered statistically significant for interaction tests of matched groups, while *p* < 0.05 was considered statistically significant for all other tests. Statistical analyses were performed using SAS statistical software version 9.4 (SAS Institute, Cary, NC, USA).

## 3. Results

### 3.1. Study Population Characteristics

In the OBSERVANT I study, 5706 patients underwent SAVR and in the OBSERVANT II study 2989 patients underwent TAVR (Figure 1). Among them, 5350 SAVR patients and 2520 TAVR patients fulfilled the prespecified criteria for the present analysis (Figure 1). The mean age of TAVR patients was 82.1 ± 6.1 years and that of SAVR patients was 73.2 ± 9.3 years (*p* < 0.001). Major comorbidities were more prevalent among patients undergoing TAVR and this translated into a markedly higher operative risk as estimated by the EuroSCORE II (7.0 ± 6.9 vs. 3.1 ± 3.9%, *p* < 0.001) compared to SAVR patients (Table 1).

Propensity score matching resulted in 1008 pairs, whose baseline risk factors were well balanced as demonstrated by an excellent overlap of the distribution of propensity score in the study groups (Appendix A) and a standardized difference < 0.10 in all covariates considered for the estimation of the propensity score (Appendix A).

Among propensity score matched pairs, mean EuroSCORE II of TAVR patients was 4.7 ± 4.0% and that of SAVR patients was 4.5 ± 5.7% (*p* = 0.419). TAVR was performed mostly using Evolut R (Medtronic Inc, Minneapolis, MN, USA) (39.0%) and Sapien 3 (Edwards Lifesciences, Irvine, CA, USA) (28.9%) devices. ACURATE neo (Boston Scientific, Marlborough, MA, USA) (12.4%), Evolut Pro (Medtronic Inc, Minneapolis, MN, USA) (11.4%), Portico (Abbott Vascular, Chicago, IL, USA) (7.4%), Lotus (Boston Scientific, Marlborough, MA, USA) (0.5%), and Engager (Medtronic Inc, Minneapolis, MN, USA) (0.4%) were the other TAVR devices employed in this series (Appendix A). TAVR was performed through a transfemoral approach in 93.2% of patients, while a transapical access site was adopted in 3.2% of patients and other peripheral access sites in 3.7% of patients (Appendix A).

### 3.2. Outcomes

Among propensity score matched pairs, TAVR was associated with a significantly lower risk of 30-day mortality (1.8 vs. 3.5%, *p* = 0.020), stroke (0.8 vs. 2.3%, *p* = 0.005), acute kidney injury (0.6 vs. 8.2%, *p* < 0.001), infectious complications (3.8 vs. 6.5%, *p* = 0.006), cardiogenic shock (1.4 vs. 5.1%, *p* < 0.001), and red blood cell transfusion (15.5 vs. 58.9%, *p* < 0.001) compared to SAVR (Table 2). However, TAVR patients had a higher risk of moderate-to-severe paravalvular regurgitation (5.9 vs. 2.0%, *p* < 0.001), permanent pacemaker implantation (during the index hospitalization, 13.8 vs. 3.3%, *p* < 0.001). The risk of major vascular complications requiring invasive treatment was significantly higher among TAVR patients as well (2.2 vs. 0.1%, *p* < 0.001) (Table 2).

At 1-year, TAVR was associated with a lower risk of all-cause death (7.9 vs. 11.5%, *p* = 0.006) (Figure 2A), MACCE (12.0 vs. 15.8%, *p* = 0.011) (Figure 2B), readmission due to heart failure (10.8 vs. 15.9%, *p* < 0.001), and stroke (3.2 vs. 5.1%, *p* = 0.033), but with a higher risk of permanent pacemaker implantation (16.2 vs. 6.4%, *p* < 0.001) compared to SAVR (Table 3).

### 3.3. Additional Analyses

Analysis of patients who underwent transfemoral TAVR showed that among 939 matched pairs TAVR was associated with significantly lower risk of mortality at one year (7.8 vs. 11.7%, HR 0.65, 95% CI 0.48–0.87) compared to SAVR patients. Subgroup analyses showed a significant interaction of age (*p* = 0.079), gender (*p* = 0.081), and EuroSCORE II (*p* = 0.072) with the treatment method on 1-year mortality (Figure 3). This effect was not observed for left ventricular ejection fraction, coronary artery disease, or diabetes. Accordingly, TAVR demonstrated its efficacy at reducing 1-year mortality in the subgroups of patients aged 80 years or older (HR 0.49, 95% CI 0.33–0.71), in females (HR 0.57, 0.38–0.85), and among patients with EuroSCORE II ≥ 4.0% (HR 0.48, 95% CI 0.32–0.71).

## 4. Discussion

In the present study, we observed that patients who underwent TAVR with new-generation devices had a rather low risk of 1-year all-cause mortality, MACCE, and hospital readmission due to heart failure, which compared favorably to propensity score-matched SAVR patients. These real-world findings are generalizable to elderly people currently undergoing invasive treatment for AS in centers with experience in TAVR.

The present analysis also showed that the risk of major vascular complications requiring invasive treatment was significantly higher among TAVR patients (2.2 vs. 0.1%, *p* < 0.001), but its incidence was numerically limited. This finding documented how advances in TAVR technology and in vascular closure devices, along with increased experience with this technique, have markedly reduced the risk of major vascular injuries, with a potential benefit for these patients. This study also documented a very low risk of stroke (0.8%) and acute kidney injury (0.6%) with TAVR in these intermediate risk patients. Instead, the incidences of moderate-to-severe paravalvular regurgitation and need of permanent pacemaker implantation were quite high and these complications seem to be the Achilles’ heel of TAVR, even with these new-generation devices. We recognize that the risk of these complications may differ according to the type of device, but a recent pooled analysis showed that the risk of permanent pacemaker implantation did not decrease with the use of newer TAVR devices (17.0% vs. 17.1%) compared to early generation devices [13]. This complication causes discomfort to the patient and increases the costs of the procedure. Still, recent studies showed that permanent pacemaker implantation after TAVR does not have any negative impact on postprocedural survival [14,15]. A pooled analysis by Mohananey et al. [14] demonstrated that permanent pacemaker implantation after TAVR had similar 1-year mortality (RR 1.03, 95% CI 0.92–1.16) and MACCEs compared to those who did not receive a permanent pacemaker. However, postoperative improvement in left ventricular function was greatest in patients without postoperative permanent pacemaker.

In this analysis, TAVR patients had a higher risk of moderate-to-severe paravalvular regurgitation compared to SAVR (5.9 vs. 2.0%, *p* < 0.001). This finding and its potential for worse survival [15,16] makes it one of the most severe adverse outcomes of TAVR. The incidence of moderate-to-severe paravalvular regurgitation in this study (5.9%) was higher than the pooled incidence of 1.8% with new-generation devices reported by Winter et al. [13]. Since the risk of paravalvular regurgitation may differ between devices [13], the development of new TAVR devices should focus on reducing the risk of this complication.

This study is a comparative analysis of the outcome of a recent prospective series of TAVR patients with a previous prospective series of SAVR patients and this may introduce bias into this analysis. Still, large nationwide studies have not documented any change in the early mortality after SAVR during the last decade [17,18]. Saad et al. [17] evaluated the outcome after aortic valve replacement in the United States from 2012 to 2017 and showed that in-hospital mortality after isolated SAVR did not decrease (range, 2.8–3.2% *p* = 0.82) during this study period. Similarly, in-hospital mortality after complex SAVR remained stable (range, 4.8–5.3%, *p* = 0.23). On the contrary, in-hospital mortality after TAVR decreased from 5.1% to 1.6% (*p* < 0.0001) (17). Consonant with these findings, Gaede et al. [18] reported the results of a German nationwide study that documented, during the years 2012–2019, that in-hospital mortality after isolated SAVR remained unchanged during this period (range 2.6–3.1%), whilst it decreased from 5.0% to 2.3% with transvascular TAVR. Therefore, we may expect that mortality rates after SAVR in all-comers have not changed significantly over time as the use of minimally invasive surgery [19] and of rapid deployment SAVR prostheses [20] failed to show any significant improvement in the outcome of these patients. A recent German multicenter study observed that the use of rapid-deployment SAVR prostheses was associated with higher in-hospital mortality compared to transfemoral TAVR (1.7 vs. 0.6%, *p* = 0.003), despite TAVR patients were significantly older than SAVR patients even after propensity score matching (median, 78 vs. 75 years, *p* < 0.001) [20]. This finding further confirmed that by virtue of the minimally invasive nature of TAVR and the advancements being made in its technology, this kind of treatment is expected to provide most benefit in the treatment of severe AS among the elderly.

In the present study, the magnitude of the effect of TAVR in reducing 1-year mortality was evident in the subgroups of patients aged 80 years or older, females, and patients with EuroSCORE II ≥ 4.0%. These findings confirmed that TAVR is most beneficial in patients with increased operative risk.

Lemor et al. [21] investigated the outcome of 84,794 patients aged >80 years who underwent aortic valve replacement (30,590 TAVR and 54,204 SAVR) from 2011 to 2015 in the United States. The authors reported lower hospital mortality after TAVR (3.4% vs. 6.8%, *p* < 0.001), with reduced risk of all major complications, shorter in-hospital stay, and reduced hospital readmission rate compared to SAVR. These results translated into decreased hospital costs after TAVR compared to SAVR (USD 59,205 vs. USD 65,146, *p* < 0.001). The benefit of TAVR among octogenarians in terms of early and mid-term survival was not evident in previous studies and, to the best of our knowledge, this issue has not been investigated in very elderly people receiving newer generation TAVR devices.

A large, pooled analysis by Panolas et al. [22] demonstrated that TAVR was associated with lower 1-year mortality among females (OR 0.68; 95% CI 0.50 to 0.94) but not among males (OR 1.09; 95% CI 0.86 to 1.39) as compared to SAVR. Similar findings were observed at 2-year follow-up. The findings have been confirmed by a more recent meta-analysis by Dagan et al. [23], and this reflects the increased operative risk of women undergoing cardiac surgery [24]. Previous studies did not document a clear benefit of TAVR in terms of survival in high-risk patients, but there is a lack of specific data in patients treated with new-generation devices.

### Limitations

The present study has several limitations, which deserve to be acknowledged. First, this a comparative analysis of patients who underwent TAVR and SAVR included in two national prospective studies. However, the lack of randomization introduces bias, which might not be completely overcome by propensity score matching. Still, propensity score matching resulted in balanced baseline covariates as shown by standardized differences < 0.10 in all risk factors considered for the estimation of the propensity score (Table 1). Preoperative hemoglobin, which was lower in TAVR patients, had a marginally high standardized difference (0.20). Second, TAVR and SAVR were performed during different study periods. Contemporary large nationwide studies did not document any change in early mortality during the last decade [17,18]. Furthermore, the observed 30-day mortality after SAVR in this series was much lower than that predicted by the EuroSCORE II (observed, 3.5%; expected 4.5%) and is comparable to the findings of recent studies [17,18]. Still, the time gap between these two series may introduce bias into this analysis. Therefore, the present study can be viewed mainly as a report of the current excellent outcome with newer generation TAVR devices in intermediate risk patients. Finally, the risk of moderate-to-severe paravalvular regurgitation and need of permanent pacemaker implantation may vary between TAVR devices. Analysis of the impact of each TAVR device on early and intermediate outcomes was not performed in this study because it was out of the scope of the present analysis. Despite the above limitations, these findings may be beneficial for clinicians because this data is from all-comers treated in centers with different referral pathways and perioperative care standards, which make these findings generalizable.

## 5. Conclusions

This study showed that elderly patients with intermediate operative risk treated with TAVR using new-generation devices had a low risk of major adverse events during the index-hospitalization and at 1-year follow-up. These results compared favorably to propensity score matched patients who underwent SAVR, particularly in octogenarians, females, and those with increased operative risk. However, the incidence of moderate-to-severe paravalvular regurgitation and need for permanent pacemaker implantation were higher in TAVR than SAVR, even with new-generation devices. Following the evidence of randomized trials, the present findings on a real-world population enforce the role of TAVR as the treatment of choice for AS in the elderly, females, and in patients with increased operative risk.

## Figures and Tables

**Figure 1 jcm-10-03703-f001:**
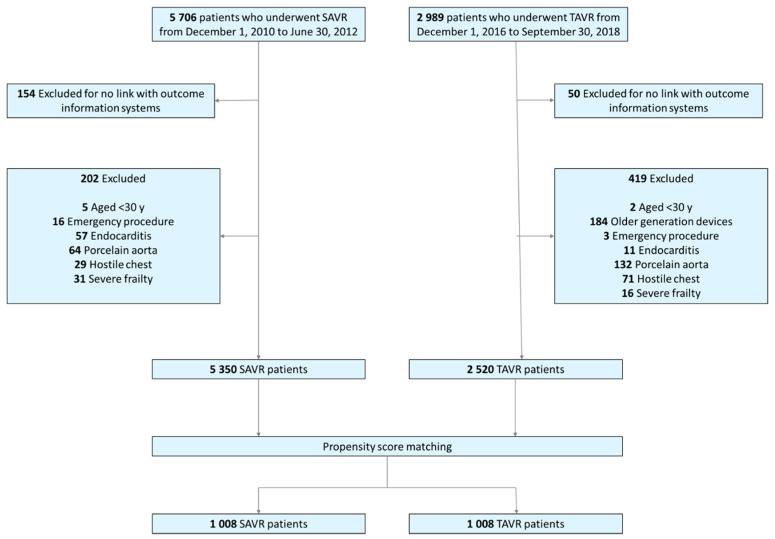
Study flowchart.

**Figure 2 jcm-10-03703-f002:**
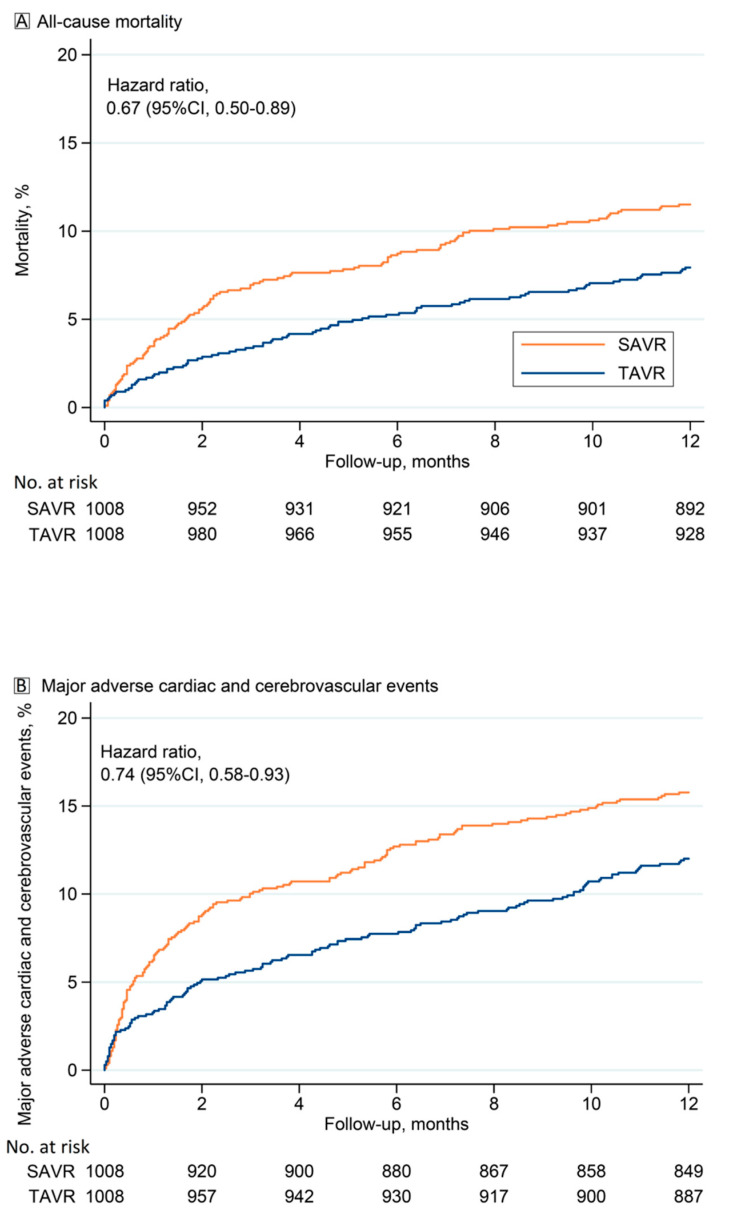
(**A**) Kaplan–Meier estimates of all-cause mortality and (**B**) of major adverse cardiac and cardiovascular events (MACCEs) after transcatheter (TAVR) and surgical aortic valve replacement (SAVR) in propensity score matched patients.

**Figure 3 jcm-10-03703-f003:**
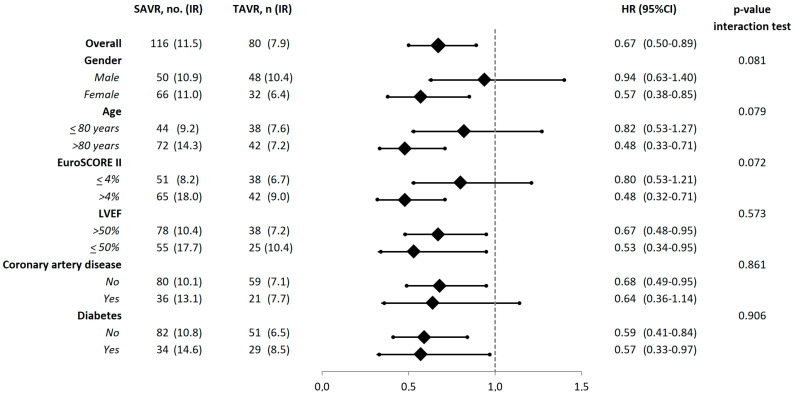
Risk estimates of 1-year mortality in subgroups of propensity score matched patients. Abbreviations: TAVR compared to SAVR; CI, confidence interval; HR, hazard ratio; IR, incidence rate; LVEF, left ventricular ejection fraction; SAVR, surgical aortic valve replacement; TAVR, transcatheter aortic valve replacement.

**Table 1 jcm-10-03703-t001:** Baseline characteristics of unmatched and propensity score of matched patients.

	Unmatched Patients			Propensity Score Matched Patients		
Variables	SAVR	TAVR	*p*-Value	SAVR	TAVR	*p*-Value
	*n* = 5350	*n* = 2520		(*n* = 1008)	(*n* = 1008)	
Age (years)	73.2 (9.3)	82.1 (6.1)	<0.001	79.3 (5.5)	79.5 (6.7)	0.357
Female	2488 (46.5)	1408 (55.9)	<0.001	567 (56.3)	557 (55.3)	0.651
EuroSCORE II (%)	3.1 (3.9)	7.0 (6.9)	<0.001	4.5 (5.7)	4.7 (4.0)	0.419
Body mass index (kg/m^2^)	27.2 (4.4)	26.4 (4.8)	<0.001	26.9 (4.7)	27.0 (4.8)	0.529
Hemoglobin (g/dL) ^a^	12.6 (1.7)	11.8 (1.7)	<0.001	12.3 (1.6)	12.0 (1.7)	<0.001
eGFR classes			<0.001			0.987
>60 mL/min/1.73 m^2^	3539 (68.0)	1143 (45.4)		563 (55.9)	572 (56.7)	
60–30 mL/min/1.73 m^2^	1441 (27.7)	1121 (44.5)		381 (37.8)	366 (36.3)	
<30 mL/min/1.73 m^2^	227 (4.4)	253 (10.1)		64 (6.3)	70 (6.9)	
Dialysis	65 (1.2)	60 (2.4)	<0.001	20 (2.0)	18 (1.8)	0.739
GSS classes			<0.001			0.895
0	4455 (83.3)	1433 (56.9)		688 (68.3)	680 (67.5)	
1	635 (11.9)	625 (24.8)		203 (20.1)	200 (19.8)	
2	260 (4.9)	461 (18.3)		117 (11.6)	128 (12.7)	
Extracardiac arteriopathy	750 (14.3)	453 (18.2)	<0.001	159 (15.8)	171 (17.0)	0.476
Pulmonary disease	516 (9.6)	389 (15.5)	<0.001	151 (15.0)	132 (13.1)	0.225
Diabetes	1246 (23.3)	694 (27.7)	<0.001	264 (26.2)	276 (27.4)	0.541
Neurological or motoric dysfunction	123 (2.3)	51 (2.0)	0.379	23 (2.3)	31 (3.1)	0.267
Liver chirrosis	96 (1.9)	40 (1.6)	0.434	29 (2.9)	27 (2.7)	0.789
Pulmonary hypertension	261 (5.2)	124 (4.9)	0.594	56 (5.6)	55 (5.5)	0.923
Active malignancy	57 (1.1)	99 (4.0)	<0.001	28 (2.8)	30 (3.0)	0.789
Oxygen therapy	49 (0.9)	81 (3.2)	<0.001	20 (2.0)	23 (2.3)	0.639
Prior aortoiliac revascularization	101 (2.0)	87 (3.5)	<0.001	28 (2.8)	31 (3.1)	0.696
Prior cardiac surgery	223 (4.2)	382 (15.2)	<0.001	66 (6.5)	76 (7.5)	0.369
Prior CABG	71 (1.3)	243 (9.6)	<0.001	33 (3.3)	42 (4.2)	0.272
Prior PCI	409 (8.0)	353 (14.0)	<0.001	111 (11.0)	117 (11.6)	0.674
Prior myocardial infarction			<0.001			0.329
No	4746 (90.4)	2168 (86.1)		906 (89.9)	885 (87.8)	
Within 90 days from the procedure	183 (3.5)	51 (2.0)		21 (2.1)	25 (2.5)	
More than 90 days from the procedure	323 (6.2)	299 (11.9)		81 (8.0)	98 (9.7)	
Critical preoperative state	84 (1.6)	51 (2.0)	0.153	19 (1.9)	18 (1.8)	0.869
NYHA classes			<0.001			0.952
1	855 (16.1)	27 (1.1)		21 (2.1)	24 (2.4)	
2	2437 (45.8)	654 (26.1)		383 (38.0)	388 (38.5)	
3	1705 (32.0)	1687 (67.5)		551 (54.7)	542 (53.8)	
4	325 (6.1)	133 (5.3)		53 (5.3)	54 (5.4)	
CCS class IV	245 (4.7)	107 (4.3)	0.408	39 (3.9)	42 (4.2)	0.732
No. of diseased coronary arteries			<0.001			0.238
0	3572 (66.8)	1841 (74.4)		765 (75.9)	756 (75.0)	
1	803 (15.0)	353 (14.3)		135 (13.4)	131 (13.0)	
2	530 (9.9)	134 (5.4)		65 (6.4)	58 (5.8)	
3	445 (8.3)	147 (5.9)		43 (4.3)	63 (6.3)	
Left ventricular ejection fraction (%)	56.5 (9.8)	53.7 (11.3)	<0.001	55.1 (10.4)	55.0 (11.0)	0.832
Left ventricular ejection fraction classes			<0.001			0.887
>50%	4130 (83.0)	1842 (73.2)		788 (78.2)	779 (77.3)	
30–50%	770 (15.5)	589 (23.4)		196 (19.4)	204 (20.2)	
<30%	74 (1.5)	87 (3.5)		24 (2.4)	25 (2.5)	
Mitral valve regurgitation			<0.001			0.980
None/trace	2528 (47.3)	367 (14.7)		251 (24.9)	248 (24.6)	
Mild	2196 (41.0)	1329 (53.2)		539 (53.5)	534 (53.0)	
Moderate	555 (10.4)	690 (27.6)		189 (18.8)	196 (19.4)	
Severe	71 (1.3)	113 (4.5)		29 (2.9)	30 (3.0)	
Aortic valve area (cm^2^) ^a^	0.7 (0.2)	0.7 (0.2)	<0.001	0.7 (0.2)	0.7 (0.2)	0.002
Aortic annulus diameter (mm) ^a^	21.7 (2.3)	22.5 (2.4)	<0.001	21.5 (2.3)	22.4 (2.3)	<0.001
Mean transvalvular gradient (mmHg) ^a^	50.7 (15.1)	47.0 (15.1)	<0.001	51.6 (14.5)	47.8 (14.2)	<0.001
Peak transvalvular gradient (mmHg) ^a^	81.7 (22.8)	75.3 (23.0)	<0.001	82.9 (21.8)	76.3 (22.2)	<0.001
Concomitant coronary revascularization	1407 (27.3)	165 (6.6)	<0.001	117 (11.6)	109 (10.8)	0.566

Continuous variables are reported as mean and standard deviation (in parentheses). Categorical variables are reported as counts and percentages (in parentheses). Abbreviations: CABG, coronary artery bypass grafting; CCS, Canadian Cardiovascular Society; eGFR, estimated glomerular filtration rate; GSS, geriatric status scale; NYHA, New York Heart Association; PCI, percutaneous coronary intervention; SAVR, surgical aortic valve replacement; TAVR, transcatheter aortic valve replacement. ^a^ variables with missing data not included in the propensity score.

**Table 2 jcm-10-03703-t002:** Early postprocedural outcomes in propensity score matched patients.

Outcomes	SAVR (*n* = 1008)	TAVR (*n* = 1008)	*p*-Value
30-day death	35 (3.5)	18 (1.8)	0.020
Stroke	23 (2.3)	8 (0.8)	0.005
Conversion to cardiac surgery	0	3 (0.3)	-
Major complication at LV apex	0	1 (0.1	-
Major vascular injury	1 (0.1)	22 (2.2)	<0.001
Acute kidney injury	81 (8.2)	6 (0.6)	<0.001
Postop. change in eGFR (mL/min/1.73 m^2^)	−12.54 (18.0)	1.9 (15.5)	<0.001
Prosthesis migration	0	13 (1.3)	-
Permanent pacemaker implantation	33 (3.3)	139 (13.8)	<0.001
Cardiogenic shock	51 (5.1)	14 (1.4)	<0.001
Infectious complication	64 (6.5)	38 (3.8)	0.006
Type of infection			0.006
Surgical site infection	20 (2.0)	4 (0.4)	
Organ/system infection	34 (3.5)	26 (2.6)	
Sepsis	10 (1.0)	7 (0.7)	
Red blood cell transfusion	580 (58.9)	156 (15.5)	<0.001
No. of transfused RBC units	1.8 (2.9)	0.3 (1.0)	<0.001
Cardiac tamponade			<0.001
Surgical treatment	35 (3.5)	6 (0.6)	
Percutaneous treatment	2 (0.2)	7 (0.7)	
Mean transvalvular gradient (mmHg)	13.5 (6.3)	8.9 (4.8)	<0.001
Peak transvalvular gradient (mmHg)	24.7 (10.7)	16.3 (7.9)	<0.001
Paravalvular regurgitation			<0.001
None/trace	840 (87.7)	559 (58.4)	
Mild	98 (10.3)	340 (35.5)	
Moderate	15 (1.6)	54 (5.6)	
Severe	5 (0.5)	5 (0.5)	
Not reported	50	50	

Continuous variables are reported as mean and standard deviation (in parentheses). Categorical variables are reported as counts and percentages (in parentheses). Abbreviations: eGFR, estimated glomerular filtration rate; SAVR, surgical aortic valve replacement; TAVR, transcatheter aortic valve replacement.

**Table 3 jcm-10-03703-t003:** One-year outcomes in propensity score matched patients.

One-Year Outcomes	SAVR (*n* = 1008)	TAVR (*n* = 1008)	*p*-Value	HR/SHR, 95% CI *
Death	11.5%	7.9%	0.006	0.67, 0.50–0.89
MACCE	15.8%	12.0%	0.011	0.74, 0.58–0.93
Readmission due to heart failure	15.9%	10.8%	<0.001	0.66, 0.52–0.85
Reoperation for aortic valve prosthesis complications	0.3%	0.4%	0.705	1.33, 0.30–5.96
Permanent pacemaker implantation	6.4%	16.2%	<0.001	2.77, 2.09–3.68
Stroke	5.1%	3.2%	0.033	0.62, 0.40–0.97
Myocardial infarction	1.8%	1.6%	0.728	0.89, 0.45–1.74
Percutaneous coronary intervention	0.2%	1.3%	0.004	6.54, 1.48–28.94
Coronary artery bypass grafting	0	0	-	-

Abbreviations: * TAVR compared to SAVR; CI, confidence interval; HR, hazard ratio; SHR, subdistributional hazard ratio; MACCE, major adverse cardiac and cerebrovascular events; SAVR, surgical aortic valve replacement; TAVR, transcatheter aortic valve replacement.

## Data Availability

The data are not publicly available due to ethical and privacy policy reasons.

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
