# Peer review of "One-Year Outcomes after Surgical versus Transcatheter Aortic Valve Replacement with Newer Generation Devices"

_jcm, 2021, doi:10.3390/jcm10163703_

Round 1

Reviewer 1 Report

The authors provide a well written manuscript comparing the periprocedural and 1-year outcome of TAVR and SAVR. Propensity score matching was used to adjust both groups for potential confounders. The study is adequately powered, with 1008 matched pairs. I hope the following comments improve the quality of the manuscript further.

  1. The results regarding the provided outcome measures seem to represent classical findings when comparing SAVR and TAVR. Can the authors please further underline or point out in detail the novelty of the study or results. I’m aware of the fact that most prospective RCT’s comparing TAVR and SAVR are influenced by specific inclusion and exclusion criteria, which drives the results. Although, the authors describe to compare an all-comers design, the novelty of the results is still not groundbreaking, or they are simply not pointed out adequately.

  1. The surgical group is a historical set of data. Comparing TAVR with an older set of patients undergoing SAVR is methodolically not sound. The authors discuss the fact that mortality following SAVR did not change / improve over time. I agree regardind this point, but how can the authors also assure that other endpoints mentioned did not change over time, and had an influence on the results. Can you please comment on that.

  1. Please consider these important points / questions regarding Propensity score matching.

  1. Why did you include patients with previous cardiac surgery. It is common knowledge that the risk of adverse events for patients with previous cardiac surgery (especially CABG with grafts crossing the midline) is higher when treated with SAVR (Same story as in PartnerII). Please prove that the inclusion of patients with previous cardiac surgery did not change your results in favour of TAVR.
  2. There are still statistically significant differences in your baseline characteristics post propensity score matching. Albumin and Hemoglobin for example, both better in TAVR-patients and both known to be related with outcome following TAVI. The standardized mean differences plot lacks to show albumin and hemoglobin, but instead refers to frailty. Can you please comment on that.
  3. Your collective post-matching also includes a certain amount of patients undergoing myocardial revascularization, but it is not clear what kind of revascularization this is in both groups.

  1. I would also consider to integrate pre-match baseline characteristics in Table 1, as it is usually done, allowing a direct comparison of both groups before and after propensity score matching. This gives a better impression of how the groups were before matching and facilitates to get an idea of which patients in each group were extracted during the matching process.

  1. Please provide detailed information of your SAVR group regarding operative details in a table, at least in the supplementary section.

  1. Please provide an explanation why almost 60% of the SAVR group needed a red blood cell transfusion, since this is surprisingly high.

  1. You stated that “Still, recent studies showed that permanent pace-maker implantation after TAVR does not have any negative impact on postprocedural survival [14,15].”: This might be true for the studies you are showing here, but there is literature clearly demonstrating that pacemaker implantation is associated with worse outcome following TAVI, especially when new pacemaker implantation occurs in combination with PVL. In summary, your statement is to general and does not reflect the entire picture.

  1. I fully agree that TAVI is an excellent solution to treat aortic stenosis, especially in elderly with intermediate surgical risk and most likely also for those with low surgical risk but only if the device landing zone is favorable for TAVI. Please integrate the entire story regarding device landing zone in your discussion, since this is the key point to allocate patients either to TAVR or to SAVR.

Author Response

Reviewer 1:

Comment 1: The results regarding the provided outcome measures seem to represent classical findings when comparing SAVR and TAVR. Can the authors please further underline or point out in detail the novelty of the study or results. I’m aware of the fact that most prospective RCT’s comparing TAVR and SAVR are influenced by specific inclusion and exclusion criteria, which drives the results. Although, the authors describe to compare an all-comers design, the novelty of the results is still not groundbreaking, or they are simply not pointed out adequately.

Answer 1: We do agree with the Reviewer on the importance to comment on the limitations of randomized clinical trials, which are the background of this study.

Changes 1: We added the following paragraph to the Introduction: “However, clinical trials might have excluded patients who still require invasive treatment for severe AS in the real-world setting. Indeed, recent studies excluded from randomization up to one third of screened AS patients [1,2]”. We have largely modified the Introduction according to these suggestions.

Comment 2: The surgical group is a historical set of data. Comparing TAVR with an older set of patients undergoing SAVR is methodolically not sound. The authors discuss the fact that mortality following SAVR did not change / improve over time. I agree regardind this point, but how can the authors also assure that other endpoints mentioned did not change over time, and had an influence on the results. Can you please comment on that.

Answer 2: We do agree with the Reviewer on this major limitation of our study. We reviewed the recent nationwide studies and summarized as well as commented their results in a rather large paragraph on this issue (see, page 11). These large studies showed “that mortality rates after SAVR in all-comers have not changed significantly over time as the use of minimally invasive surgery [19] and of rapid deployment SAVR prostheses [20] failed to show any significant improvement in the outcome of these patients”. We believe that these studies provided solid data on the lack of any significant improvement of SAVR in these patients.

Changes 2: None.

Comment 3: Why did you include patients with previous cardiac surgery. It is common knowledge that the risk of adverse events for patients with previous cardiac surgery (especially CABG with grafts crossing the midline) is higher when treated with SAVR (Same story as in PartnerII). Please prove that the inclusion of patients with previous cardiac surgery did not change your results in favour of TAVR.

Answer 3: We respectfully disagree with the Reviewer on this issue. Surgery on the aortic valve prosthesis in patients with prior cardiac surgery is still indicated when anatomical characteristics of the aortic root and the aortic valve prosthesis does not allow valve-in-valve TAVR. Furthermore, these patients belong to the all-comers population and we believe they should be included in this analysis. We did not perform interaction test of this subgroup of patients because their small number prevent reliable results.

Changes 3: None.

Comment 4: There are still statistically significant differences in your baseline characteristics post propensity score matching. Albumin and Hemoglobin for example, both better in TAVR-patients and both known to be related with outcome following TAVI. The standardized mean differences plot lacks to show albumin and hemoglobin, but instead refers to frailty. Can you please comment on that.

Answer 4: We thank the Reviewer for such a detailed evaluation of our data. Indeed, we reported data on preoperative albumin and hemoglobin, but these covariates were not included in the regression model for estimation of the propensity score because of missing data in a large number of patients (355 patients in each matched cohort). Including these covariates would have led to exclusion of a large number of patients. It is worth noting that albumin had the largest amount of missing data and therefore, its mean values are not reliable in this context. Therefore, we decided to delete data on albumin from the new Table 1. We kept the hemoglobin data in the manuscript. Please note that preoperative hemoglobin was lower in the TAVR cohort, which is not in favor on these patients. However, the estimated standardized difference after matching was 0.20, which is not excessively high. It is worth noting that the study cohorts were well matched for a large number of other significant comorbidities. 

Changes 4: We added this information to the Methods section, Table 1 and Suppl. Table 2. We commented on this finding in the Limitations section.

Comment 5: Your collective post-matching also includes a certain amount of patients undergoing myocardial revascularization, but it is not clear what kind of revascularization this is in both groups.

Answer 5: The OBSERVANT registries did not include specific data on the type of coronary revascularization performed in these patients.

Changes 5: None.

Comment 6: I would also consider to integrate pre-match baseline characteristics in Table 1, as it is usually done, allowing a direct comparison of both groups before and after propensity score matching. This gives a better impression of how the groups were before matching and facilitates to get an idea of which patients in each group were extracted during the matching process.

Answer 6: We do agree with the Reviewer on the need to merge these tables.

Changes 6: We merged these tables into a new Table 1 as kindly suggested.

Comment 7: Please provide detailed information of your SAVR group regarding operative details in a table, at least in the supplementary section.

Answer 7: The OBSERVANT registries did not include specific data on operative details of SAVR.

Changes 7: None.

Comment 8: Please provide an explanation why almost 60% of the SAVR group needed a red blood cell transfusion, since this is surprisingly high.

Answer 8: We thank the Reviewer for this comment. Patients undergoing SAVR have an increased need of blood transfusion compared to TAVR patients. The rather high rate of blood transfusion after SAVR observed in this nationwide study has been reported also in previous large studies. In a recent Finnish nationwide study on 4333 SAVR patients, 70% of patients received blood transfusion after surgery (https://pubmed.ncbi.nlm.nih.gov/31112060/).

Changes 8: None.

Comment 9: You stated that “Still, recent studies showed that permanent pace-maker implantation after TAVR does not have any negative impact on postprocedural survival [14,15].”: This might be true for the studies you are showing here, but there is literature clearly demonstrating that pacemaker implantation is associated with worse outcome following TAVI, especially when new pacemaker implantation occurs in combination with PVL. In summary, your statement is to general and does not reflect the entire picture.

Answer 9: We thank the review for this comment. Meta-analyses showed that implantation of a permanent pace-maker does not impact the survival of TAVR patients. We reported the main findings of a large meta-analysis by Mohananey et al. (14). A more recent large multicenter study confirmed these findings (Chamandi et al. https://pubmed.ncbi.nlm.nih.gov/31129090/).

Change 9: We further described the results of the meta-analysis by Mohananey et al. in the Discussion.

Comment 10: I fully agree that TAVI is an excellent solution to treat aortic stenosis, especially in elderly with intermediate surgical risk and most likely also for those with low surgical risk but only if the device landing zone is favorable for TAVI. Please integrate the entire story regarding device landing zone in your discussion, since this is the key point to allocate patients either to TAVR or to SAVR.

Answer 10: The introduction of TAVR technology has widened the indications/feasibility of TAVR in either low- and high-risk patients. For example, a large number of patients with stenotic bicuspid valve is currently treated with TAVR. We recognize that anatomical characteristics contraindicating TAVR still exist, but allocation to TAVR or SAVR is mostly based on patient’s age and comorbidities. We believe that the discussion of these aspects are somewhat out of the topic of the present study.

Change 10: None.

Reviewer 2 Report

Dear editor and authors,

This is an interesting study comparing SAVR and TAVR with new generation valves in all-comers. I would like to congratulate the authors for the inclusion of more than 1000 patients in 2 propensity-matched cohorts, especially with new generations percutaneous valves.

I still do have some comments and suggestions.

Introduction

My major concern is that the purpose of the study is somewhat unclear: the authors begins by mentioning good TAVR results in low risk patients and in bicuspids while bicuspid are not developed later in the manuscript, and only thereafter are high-risk patients mentioned and their good response to SAVR. It seems here that the authors focus on this surgical risk strata, while the aim of the study appears to be related to all comers.

The author should clarify if and why they focus on this subgroup, especially while TAVI is already a class 1 recommendation in this population. I suggest to mention rapidly the current guidelines recommendations. Also, the authors should better introduce the state-of-the-art regarding TAVR versus SAVR results, by mentioning current RCTs as well as studies in all-comers such as GARY registry or the study of Brennan et al. (J Am Coll Cardiol 2017;70:439-50) and what is missing in those trials.

Also the main objective is to compare TAVR with new generation devices to SAVR, but this has not been introduced clearly.

Material and methods

Please describe the method used for propensity matching (propensity score description, choice of covariates).

Results :

-Supp fig 1 : what does the colored curves refer to ?

-Supp fig 2 : provide legend for color use

-Supp fig 3 : Correct IR values for age subgroups (confusion between mortality and survival rates)

-Cite figure 2 (Kaplan-Meier curves) and add p values

-I suggest to combine tables 2 and 3 and move Suppl. fig 3 to general figures.

Do the authors have information about the role of patient anatomy (arteries, aortic annulus shape) ?

Discussion

The results show an interaction between TAVR superiority and age, sex and surgical risk, more precisely TAVR appears superior to SAVR in the specific subgroups of patients >80 years, in women and in the high risk group. Could the authors provide some explanation for these findings and confront the results to other studies. Particularly, does it challenge current studies in low risk and/or younger patients?

It could be interesting to detail the surgical devices that were implanted (type such as sutureless, stentless … and sizes) and provide information on the rate of prosthesis-patient mismatch. Could there be a balance between favorable antegrade hemodynamics and deleterious leaks in TAVR patients explaining the results ? In that sense, is the benefit noted in TAVR rather due to new generation devices in TAVR or older generation of devices in SAVR ?

Conclusion

Finally, what should drive the decision to perform SAVR or TAVR ? (surgical risk ? age ? sex ? else ? (ex : clinical status ? anatomy, procedural factors…)

Author Response

Reviewer 2:

Comment 1: My major concern is that the purpose of the study is somewhat unclear: the authors begins by mentioning good TAVR results in low risk patients and in bicuspids while bicuspid are not developed later in the manuscript, and only thereafter are high-risk patients mentioned and their good response to SAVR. It seems here that the authors focus on this surgical risk strata, while the aim of the study appears to be related to all comers. The author should clarify if and why they focus on this subgroup, especially while TAVI is already a class 1 recommendation in this population. I suggest to mention rapidly the current guidelines recommendations. Also, the authors should better introduce the state-of-the-art regarding TAVR versus SAVR results, by mentioning current RCTs as well as studies in all-comers such as GARY registry or the study of Brennan et al. (J Am Coll Cardiol 2017;70:439-50) and what is missing in those trials. Also the main objective is to compare TAVR with new generation devices to SAVR, but this has not been introduced clearly.

Answer 1: We do agree with the Reviewer that the aim of this study was to investigate the outcome of all-comers, particularly when newer generation TAVR were employed. We have revised the Introduction commenting the findings of the study by Brennan et al.’s and clarifying the background of the present study. We believe it is not necessary to summarize the results of recent randomized trials because they are well among cardiologists and cardiac surgeons.

Changes 1: We added the following paragraphs to the Introduction: “Randomized clinical trials provided evidence on the early and mid-term efficacy and safety of TAVR in AS patients within a broad spectrum of operative risk. However, clinical trials might have excluded patients who still require invasive treatment for severe AS in the real-world setting. Indeed, recent studies excluded from randomization up to one third of screened AS patients [1,2].” and “These comparative analyses often included patients who received older generation TAVR devices, which prevented a reliable assessment of the outcome with newer TAVR devices.”.

Comment 2: Please describe the method used for propensity matching (propensity score description, choice of covariates).

Answer 2: We reported the details regarding the methods employed for propensity score matching analysis herein employed. The choice of covariates was based on a non-parsimonious approach, therefore including all covariates available in the registries. We added information on the covariates excluded because of missing data.

Changes 2: We added a few details on the methods employed for propensity score matching.

Comment 3: Supp fig 1 : what does the colored curves refer to ? Supp fig 2 : provide legend for color use. Supp fig 3 : Correct IR values for age subgroups (confusion between mortality and survival rates). Cite figure 2 (Kaplan-Meier curves) and add p values. I suggest to combine tables 2 and 3 and move Suppl. fig 3 to general figures.

Answer 3: We do agree with the Reviewer on the need of these changes. Regarding Log-rank test’s p-values, they are reported in text. We prefer to keep tables 2 and 3 as they are. In fact Table 3 include HRs and SHRs and readers may be most interested in 1-year results.

Changes 3: We modified supplemental figures’ legends, corrected IRs, moved suppl. fig. 3 to the main manuscript (new figure 3) and cited Kaplan-Meier’s curves to the Results.

Comment 4: Do the authors have information about the role of patient anatomy (arteries, aortic annulus shape) ?

Answer 4: The OBSERVANT registries do not include such information.

Changes 4: None.

Comment 5: he results show an interaction between TAVR superiority and age, sex and surgical risk, more precisely TAVR appears superior to SAVR in the specific subgroups of patients >80 years, in women and in the high risk group. Could the authors provide some explanation for these findings and confront the results to other studies. Particularly, does it challenge current studies in low risk and/or younger patients?

Answer 5: The present results are somewhat new, at least regarding the potential benefit of TAVR in octogenarians and high risk patients. The are not many studies which assessed these issues, particularly in patients receiving newer TAVR devices. The benefits of TAVF in women has been previously documented in at least two meta-analyses.

Changes 5: We reported the results of the literature on these issues to the Discussion.

Comment 6: It could be interesting to detail the surgical devices that were implanted (type such as sutureless, stentless … and sizes) and provide information on the rate of prosthesis-patient mismatch. Could there be a balance between favorable antegrade hemodynamics and deleterious leaks in TAVR patients explaining the results ? In that sense, is the benefit noted in TAVR rather due to new generation devices in TAVR or older generation of devices in SAVR ?

Answer 6: The OBSERVANT registries do not include information on the type of surgical aortic valve prostheses. Therefore, we are unable to perform such relevant analyses.

Changes 6: None.

Comment 7: Finally, what should drive the decision to perform SAVR or TAVR ? (surgical risk ? age ? sex ? else ? (ex : clinical status ? anatomy, procedural factors…)

Answer 7: The present study showed that octogenarians, females and high-risk patients may have improved 1-year survival after TAVR compared to SAVR.

Changes 7: This message was included in the Conclusions. These results were also added to the Results in the Abstract

Round 2

Reviewer 1 Report

Congratulations to the revised manuscript. The quality of the manuscript significantly improved throughout the revision process.